# Non-Destructive Detection of Moldy Walnuts Based on Hyperspectral Imaging Technology

**DOI:** 10.3390/molecules27206776

**Published:** 2022-10-11

**Authors:** Junyan Xu, Daochun Xu, Xiaopeng Bai, Rongchao Yang, Jiale Cao

**Affiliations:** 1Key Lab of State Forestry and Grassland Administration for Forestry Equipment and Automation, School of Technology, Beijing Forestry University, No. 35 Tsinghua East Road, Haidian District, Beijing 100083, China; 2Agricultural Information Institute, Chinese Academy of Agricultural Sciences, Beijing 100081, China

**Keywords:** moldy walnut, hyperspectral imaging technology, classification, non-destructive

## Abstract

Walnuts with their shells are a popular agricultural product in China. However, mildew from growth can sometimes be processed into foods. It is difficult to visually determine which walnuts have mildew without breaking the shells. A non-destructive method for detecting walnuts with mildew was studied by combining spectral data with image information. A total of 120 “Lüling” walnuts with shells were used for the mildew experiment. The characteristics of the spectral data from six surfaces of all samples were collected in the range of 370–1042 nm on days 0, 15, and 30. The spectrum was pretreated using SNV, and the feature bands were extracted using PCA and modeled using a support vector machine (SVM). The results show that the overall classification accuracy was 93%, with an of accuracy of 100% for INEN walnuts (normal internally and externally). The accuracy for IMEM walnuts (mildew internally and externally) reached 87.29%. There was an accuracy of 78.6% for IMEN walnuts (mildew internally and normal externally). The non-destructive detection of mildewed walnuts can be undertaken using hyperspectral imaging technology, which provides a new technique for exploring the mechanisms of walnuts with mildew.

## 1. Introduction

Walnut (*Juglans regia* L.) is an important woody oilseed tree species in China, with a high status in terms of economic and nutritional value. Walnuts contain a balanced set of three nutrients. Among them, the fat content is approximately 50–60% and the protein is 15–24%, with 12–16% carbohydrates, in addition to essential trace elements and vitamins [1,2,3]. Walnuts can strengthen the brain, protect the liver, regulate cardiac metabolic diseases, and reduce the risk of other non-communicable diseases [4].

China has the most extensive area for the cultivation and output of walnuts worldwide as well as has the highest walnut consumption [5]. With their high nutritional value, walnut kernels can be eaten alone or can be made into other products such as walnut milk, oil, pastries, and candy. In recent years, with continuous improvements to quality of life, consumer demand for processed walnut products is also growing. The quality of walnuts plays a decisive role in these purchases, which puts forward strict requirements for China’s walnut processing production lines.

Walnuts with shells are popular products in China. After ripe walnuts have been picked, during transportation, processing, and storage, problems such as oil oxidation, protein deterioration, color browning, flavor loss, and seed kernel mildew can occur [6,7]. When problem walnuts are eaten by mistake, it can cause mild food poisoning. Moldy walnuts contain a lot of toxins, and severe cases can even cause cancer. At present, the traditional detection method used to identify walnuts that have been harmed is manual detection. During the sampling of walnut batches, it becomes necessary to break the shell to determine its internal quality. Physical means or stoichiometry can be used to examine the walnut fatty acid content, oil oxidation value, seed kernel color, and other key values. However, this method is time-consuming and labor-intensive and destroys the integrity of the walnut shells [8].

There are several non-destructive detection methods that have been proposed by researchers to evaluate in-shell walnuts. Jiang et al. [9] combined electronic nose responses with physicochemical methods to detect internal quality changes in Chinese pecans after different storage times. The voting method performed better in classification, with a 100% accuracy rate. A partial least squares regression (PLSR) was used for the quantitative regression of the acid values and peroxide values of pecan kernels. This study found that E-nose had a high correlation with the internal quality of pecans. However, during application, different detectors and detection environments have a greater impact on the response signal of E-nose, and it is essential for the build models that can be adjusted for different situations. X-ray imaging technology has also been applied in the non-destructive testing of walnuts. Researchers examined the internal qualities of hard-shelled walnuts, such as the integrity of the kernel, including whether they have been mothed by insects or become hollow and the size of the walnut kernel [10,11]. Bernard et al. reconstructed 3D images of walnuts using X-ray computed tomography (CT) and obtained morphological measurements of walnuts, including the length, face, and peel diameter, and characteristics that previously required the opening of walnuts, including the shell thickness, kernel volume, and nut/shell ratio. Traits were determined that were previously difficult to quantify, including the shell roughness, nut spherical degree, nut surface area, and nut shape. However, the technique was expensive and failed in obtaining the qualitative transformation characteristics of walnuts [12]. The application of spectroscopic technology in non-destructive nut testing has also become increasingly extensive. Terahertz time domain spectroscopy (THz-TDS) is a non-contact measurement technique in which the THz spectrogram characterizes the interactions between molecules, enabling the measurement of physical information about the sample [13]. Qi et al. [14] collected the THz-TDS spectra of insect borer, mildew, and normal walnut shells and kernel slice standards. The differences between the metamorphic and normal walnut spectra and absorption spectra were then compared and analyzed. Bin et al. [15] measured the terahertz band absorption curve in normal and insect-bored American pecan slices. This laid the foundation for the exclusion of metamorphic walnuts. Near-infrared spectroscopy (NIR) can react to the main chemical bond vibrations in organic molecules. Some researchers have used mapping combined with stoichiometry to measure the water, fat, protein, and other contents as well as antioxidant characteristics in walnuts. However, the near-infrared spectral resolution is high, but its point scanning method can only provide an average spectrum of the object to be detected, and it is unable to reflect the spatial distribution state of the analyte [16,17,18,19].

Hyperspectral imaging can make up for the shortcomings of near-infrared spectroscopy by combining images with near-infrared spectroscopy to obtain three-dimensional optical image data comprising a range of wavelengths [20]. Ma et al. [21] collected hyperspectral images of “Wen 185” walnut kernels from Xinjiang in the ranges of 862.9–1704.02 nm and 382.19–1026.66 nm and established protein and fat content prediction models, respectively. This encompassed the full spectral band and the characteristic spectral band using the partial least squares regression (PLSR) algorithm. A classification model of walnut appearance quality was then established using the decision tree algorithm. Chen et al. [22] performed PLS modeling and analysis on data processed using different spectral pretreatment methods. They concluded that modeling after SUV treatment was the most effective technique for sorting walnut shells, walnut kernels, and distracted wood. Nogales-Bueno et al. [23] applied hyperspectral imaging techniques to identify five species of walnuts with 96% accuracy.

Hyperspectral imaging technology is favored by researchers for the non-destructive testing of nuts because of its perfect spectral binding properties. Despite this, there are a few researchers that are committed to the field of examining intact walnuts with shells. This is because the hard shells of walnuts make it relatively difficult to study. To date, no researcher has applied this technology to the non-destructive testing of potentially moldy walnuts. This study is based on hyperspectral imaging technology to explore non-destructive testing methods for intact moldy walnuts with shells.

## 2. Results and Discussion

### 2.1. Comparison of Classification Models Based on Original Spectral Data

After any abnormal samples had been removed using the Martens distance discriminant method, walnuts with mildew (M) or that were normal (N) internally (I) and externally (E) were combined, and the spectral data of the back and bottom surfaces were input as variables. Mildewed walnuts were marked as 1, and walnuts that were normal were marked as 0. The results were divided into four categories, namely IMEM, INEM, IMEN, INEN. After manual discrimination, the numbers of samples in these four categories were 55, 1, 20, and 36. Three classification models were built based on the spectral data.

#### 2.1.1. K-Nearest Neighbor (KNN)

The traditional K-nearest neighbor algorithm is a classic machine learning algorithm proposed by Cover and Hart [24]. The basic principles are to calculate the distance between the sample to be classified and all learning samples. The closest k of the nearest neighbors is then selected and classified. KNN usually outputs most classes of the k nearest neighbors and is sensitive to the k value. After comparing the experimental results, K = 4 in this paper.

#### 2.1.2. Decision Tree (DT)

Decision trees are one of the most common techniques in data mining. They initially appeared in the CLS (Concept Learning System). A model can be learned from a given training dataset from a decision tree to classify unpredictable examples to obtain the final decision result [25]. A decision tree comprises a root node, several internal nodes, and several leaf nodes [26]. The Gini coefficient served as a dividing criterion in this study.

#### 2.1.3. Support Vector Machine (SVM)

The SVM algorithm is designed for classification and regression problems. It aims to construct an optimal separating plane (SP) by mapping the original data points from the input space to form a high-dimensional feature space. The distances from all data points to the SP are the minimum. The kernel function performs nonlinear mapping and plays an important role in the procedure [27]. SVM can be used for linear and nonlinear multivariate analysis problems, using systems of linear equations instead of quadratic programming to solve support vectors. By selecting the appropriate kernel function, the speed and efficiency of the modeling are guaranteed while generating a nonlinear map [28]. The present study used RBF kernels for training.

Three classification models were built based on full-band spectral data, with the training and verification sets randomly assigned in a 4:1 ratio, with each classifier operating 10 times to calculate its average value. The overall classification accuracy (OA) results are shown in Table 1.

Table 1 provides data on the overall classification accuracy (OA) results. Among the three classification models, the accuracy of the spectral data test collected on day 15 was generally higher than on day 30. However, the overall classification accuracy for the different mildew conditions did not exceed 90%. The OA for the selection of the walnut back surface to establish a classification model was slightly higher than for the bottom surface, and the SVM was better than the other models.

Given that the main purpose of this study was to explore whether hyperspectral imaging technology can successfully identify mildewed walnuts, especially for IMEN walnuts, it is necessary to consider the accuracy of the classification of the various mildew conditions. The accuracy of each category (CA) is shown in Table 2.

According to Table 2, SVM had the strongest performance in the classification of INEN walnuts once it reaches 100%. However, its classification of IMEN walnuts was less effective. However, KNN and DT had pronounced effects on this variety of walnut. With the back surface hyperspectral data of 30-day walnut samples as the input, the accuracy of the DT classifier for IMEN walnuts reached 38.33%. The main reason for this result is that there were relatively few samples in this category and the training and verification set were randomly divided. It is relatively difficult to distinguish the characteristics of this variety of walnut, which may be divided into the training set. This also indicates that the KNN and DT classifiers are expected to recognize this kind of walnut. DT was found to be better for the identification of moldy walnuts based on full-band spectral data, and SVM was better for the classification of intact walnuts. Given that the classification accuracy of moldy walnuts was less than 90%, further analysis of the spectral data was needed.

### 2.2. Spectral Data Preprocessing

The accuracy of the classification model established based on the original spectral data was not high. There were also many noise signals in the spectrum, which could interfere with the analysis process and directly affect the model accuracy. It was necessary to preprocess the original hyperspectral data to further explore its rules. In this study, SG convolution smoothing (SG), first-order differential (FD), second-order differential (SD), multiple scattering correction (MSC), standard normal transformation (SNV), SG-MSC, SG-FD, and SG-MSC-FD were compared to the original spectral data [29].

To compare the advantages and disadvantages of these pretreatment methods, three identification methods of each type of preprocessing data model were used to establish the classification of walnuts with mildew. However, only the back side hyperspectral walnut data on days 0 and 30, with a total of 232 (120 + 112) measurements, were used as the input, and the classification results are shown in Figure 1.

As shown in Figure 1, the OA of the hyperspectral data modeling based on KNN and the SVM algorithm improved compared with that of the original data after preprocessing. However, the OA and CA of the DT algorithm modeling both decreased. The SVM algorithm still could not identify IMEN walnuts. SG, SD, SG-MSC, and SG-FD showed similar modeling effects, with an average accuracy of 80.5%, and FD, MSC, SG-MSC-FD, and SNV achieved 82%. The FD and SNV preprocessing methods had balanced OA among the three modeling algorithms. Therefore, the subsequent research on hyperspectral band extraction and the establishment of the walnut classification model are based on FD and SNV preprocessing hyperspectral data.

### 2.3. Extraction of Feature Bands

Full-wavelength hyperspectral imaging generates a considerable amount of data, so it leads to extensive analysis to directly use them for the calculation. There is also a lot of noise and redundant information in the full-wavelength data. This interferes with the information contained in the spectrum of walnut samples, which affects the judgment accuracy. Therefore, selecting the most effective spectral information in the entire band can improve the classification accuracy and simplify the calculation. In this study, the supervised feature band selection method was used to compare hyperspectral data preprocessed by FD and SNV with the successive projections algorithm (SPA), correlation coefficient (CC), and PCA.

#### 2.3.1. Successive Projections Algorithm (SPA)

In view of these research findings, the samples’ back surface hyperspectral data preprocessing on days 0,15, and 30 and characteristic wavelengths were extracted as input variables to perform the modeling operations. OA and CA were also taken as evaluation indices. Given that the sample number of the category for INEM walnuts was too small, there was no analytical value, and the results are shown in Figure 2.

In Figure 2a, the modeling effect of extracting characteristic bands by SPA after the spectral data had been preprocessed by SNV was more effective. The OA of the three modeling methods was higher than 80%. Figure 2b represents the CA of the IMEM walnuts. Figure 2c shows that the overall modeling effect of the classification of IMEN walnuts was relatively poor and that the classification accuracy was less than 20%. Figure 2d shows that the CA of the classification of INEN walnuts through SVM can reach 100%, but also shows that the three modeling methods can misjudge mildewed walnuts as being normal walnuts.

#### 2.3.2. Correlation Coefficient (CC)

The correlation coefficient was undertaken using the same process, except CC was the method used to extract characteristic wavelengths. The results are shown in Figure 3.

In Figure 3a, the OA of the spectral data preprocessed by SNV and feature bands extracted by CC reached up to 86.5%. Figure 3b represents the CA of the IMEM walnuts and demonstrates that the decision tree modeling method is effective in distinguishing mildewed walnuts. However, in Figure 3c, the DT performance for IMEN walnuts was 10%, indicating that the two spectral pretreatment methods and feature band extraction are not conducive to this classification. In Figure 3d, the classification accuracy for INEN walnuts could also reach 100% after SVM.

#### 2.3.3. PCA

PCA can effectively screen spectral features with high contribution rates and showed that the contribution rates of the principal components in the first ten dimensions reached 100%. Therefore, the feature bands for the first ten dimensions were selected for subsequent modeling operations. The modeling method was the same, and the classification is shown in Figure 4.

Figure 4a shows that the OA of the spectral data obtained on day 15 by the SNV-PCA-SVM combined modeling method reached 93%. Figure 4b shows that the accuracy of the classification of IMEM walnuts reached 87.29% under the combined modeling of SNV-PCA-SVM. However, this combination performed poorly in the discrimination of the CA of IMEN walnuts, as seen in Figure 4c. In Figure 4d, the discrimination accuracy for INEN walnuts is 100%.

### 2.4. Analysis of Results

According to the results, the accuracy for IMEN walnuts was 38.33% when a DT base was built onto the original spectrum data, and it was reduced when the spectrum was modeled by SNV-PCA-SVM. It is difficult to distinguish IMEN walnuts from INEN walnuts only using the back surface, and almost all IMEN walnuts were misjudged as INEN walnuts. Therefore, the method requires the identification of IMEN walnuts from INEN walnuts as a second process.

Three classification models were built based on a mix of the two original spectral datasets from 20 IMEN walnuts and 156 INEN walnuts. The training set and verification set were randomly assigned in a 4:1 ratio, with each classifier operating 10 times. The classification accuracy is shown in Figure 5.

Figure 5 shows that the accuracy for IMEN walnuts was 78.6% when DT was built based on the original bottom surface spectral data. Considering the penetration of the spectrum and the bottom features of the walnut, the mould first attached to the surfaces of the walnuts when they were shelled (without cracking) and then spread inside from the hole in the bottom at the junction of the walnut and peduncle. Given that there is little literature on mildew in walnuts, it is likely that this is also the reason why “Wenwan” walnuts have seal wax at the base.

Overall, our studies focused on walnut mold regularity and a preliminary exploration of the non-destructive detection of moldy walnuts based on hyperspectral imaging technology and establish the SNV-PCA-SVM model to classify moldy walnuts with an accuracy of 87.29% for IMEM walnuts using the back surface. Then, the DT was established based on the bottom surface original spectrum, with a CA of 78.6% for IMEN walnuts in the second process. Our study serves as a proof of concept that the non-destructive detection of moldy walnuts by obtaining hyperspectral imaging on the different surfaces of walnuts is feasible.

Although there are important discoveries revealed by this study, there are also limitations. First, the sizes of the four categories of walnuts should be larger. However, because it was difficult to obtain IMEN walnuts with their shells unbroken in nature, the cultivation of moldy walnuts in the present study must be carried out before the hyperspectral imaging data acquisition. Therefore, it is also important to study the regularity of mold growth in walnuts for further research in order to obtain more IMEN walnut samples. Second, the algorithm needs to be optimized in the future to improve the accuracy.

## 3. Materials and Methods

### 3.1. Materials

“Lüling” walnuts were harvested in September 2020 from the Walnut Demonstration Base of Hebei Lüling Fruit Co., Ltd. The walnuts were peeled, dried naturally, and stored in cold storage (2–5 °C) for three months. Among them, 120 walnuts with large sizes, smooth shell surfaces, and good quality of appearance were randomly selected as test materials. These walnuts were then divided into three groups, and each nut was marked. For the walnuts to grow mildew in the naturally humid environment, they were treated with different levels of water content.

### 3.2. Hyperspectral Data Acquisition

Hyperspectral data of the walnut samples were obtained using a hyperspectral imaging system (Beijing Anzhou Technology Co., Ltd., Beijing, China). The system comprised an SOC710VP portable hyperspectral imaging spectrometer, a halogen light source, a storage platform, and a computer. Its spectral range was 370–1042 nm, and the spectral resolution was 4.6875 nm, with 128 spectral bands and 696 × 520 pixels. There was a built-in panning scrubbing device in the system. The scanning speed and integration time were automatically matched without manual adjustment, which meant that this system could avoid image distortion.

During walnut mildew growth, the hyperspectral data of the six surfaces (front, back, top, bottom, lateral1, and lateral2) were obtained for each walnut sample on days 0, 15, and 30. During the experiment, each walnut was placed directly below the spectrometer lens, and data on the six sides were collected facing up. After comparison, the exposure time was set at 6.5 ms, and the hyperspectral images obtained from the sample with the 35 cm lens were the clearest. To avoid the influence of external light sources, the hyperspectral imaging system was surrounded with a black shading cloth. This was undertaken to ensure that the test environment was not disturbed by other light, as shown in Figure 6.

To eliminate uneven distribution of the intensity of the light source in each band and the influence of the noise generated by the dark current in the camera on the spectral data, the original images were black and white corrected. The acquisition of the hyperspectral images and the black and white correction process were completed using the acquisition software supporting the instrument, and the correction formula was as follows:(1)$$I=\frac{I_{O}-I_{D}}{I_{D}-I_{W}}$$
where I is the corrected spectral image file, $I_{0}$ is the raw spectral image file, $I_{D}$ is the black image file, and $I_{W}$ is the white image file.

After all the hyperspectral images of the walnuts had been collected on day 30, 120 walnuts were opened to observe their internal conditions. The results were input as variables for the subsequent building of the discriminant model.

### 3.3. Analysis of Hyperspectral Data

Each walnut hyperspectral image was cropped into 168 mm (W) × 184 mm (H) in ENVI5.1 (ITT Visual Information Solutions Boulder, CO, USA), and the wavelength range was intercepted to 400–1000 nm to avoid noise interference. Then, the region of interest (ROI) was extracted and established to form a mask image. The hyperspectral images collected for each sample on days 0, 15, and 30 were stitched together in chronological order according to pixels to form a combined image, and PCA was then performed. From the results, the contribution rate of the first three principal components exceeded 99.90%, and the analysis results from the other surfaces were found to be similar.

Figure 7a shows the original image of the back surface (day 0–day 15–day 30) of a single walnut, and the PC1–PC3 score plots are illustrated in Figure 7b. PC1 predominantly reflects common characteristics such as the gray value of the entire walnut. PC2 is mainly manifested by differences in the color of walnut skin. The color of the walnut shell is light yellow on day 0 then becomes darker, and the absorbance is enhanced on days 15 and 30 due to changes in the water content. PC3 predominantly shows the texture of the walnut surface and the potential presence of foreign bodies. If there is mildew growing on the surface of a walnut, the mildew and the normal walnut can be distinguished by performing a PCA on the image. However, many walnuts have had mildew growing internally with no pronounced changes on the outside of the nut. The sample in Figure 7 was extracted and determined using a DNA sequence contrast. Two pathogenic bacteria were isolated, namely *Aspergillus niger* and *Aspergillus aflex*, as shown in Figure 7c.

Given that the walnuts have a clear boundary with the background at the 533 nm band in the original image, the entire walnut region at this band was regarded as an ROI to extract the spectral data in MATLAB R2012a (MathWorks, R2012a, U.S.). The process flow is shown in Figure 8. The reflected spectra from all pixels on the walnut at each band were extracted, and the results are shown in Figure 9.

As shown in Figure 9a, after extracting the entire walnut as the region of interest, the spectral trends of walnuts on day 0 and day 30 after mildew treatment were found to be the same. The main difference was that the reflectivity below 700 nm of the majority of spectra acquired on day 30 was lower than in the data acquired on day 0. This was predominantly due to the color change of the shell from mold growth. However, the reflectivity at 700–1000 nm of some samples was higher at day 30 than in normal walnuts. This band belongs to the near-infrared. The average spectral trend for all normal (0 day) walnuts with six surfaces is shown in Figure 9c. The average reflectance of the top and bottom walnut surfaces was lower than the other four surfaces. This was caused by the size of the area of interest. The spectral reflectance of individual walnut samples on day 0, 15, and 30 varied substantially. However, the determination of the presence of mildew inside the walnut shell required further modeling and analysis. According to the degree of dispersion and the reflectance trend analysis for the spectral data of all mildew and normal walnuts with six surfaces, the back and bottom surfaces were selected for the subsequent pretreatment and modeling analysis. Because the high and low ends of the original spectrum had a lot of random noise, only 115 bands of reflected spectral data in the 400–1000 nm range were intercepted in this study [21].

## 4. Conclusions

Hyperspectral imaging technology was used to explore non-destructive mildew testing methods in walnuts. The spectral data characteristics of six surfaces of walnut samples with a total of 2160 (120 × 6 × 3) measurements before and after mildew were obtained. Through the mildew experiment with walnuts with shells, four results could be produced due to the environment and environmental conditions during mildew growth, namely IMEM, INEM, IMEN, INEN.

Through a combination of spectral and image information, when the spectra were modeled using SNV-PCA-SVM, the overall classification accuracy could was 93%, in which the CA of INEN walnuts was 100%. The CA of IMEM walnuts was 87.29%, indicating that this pretreatment method can effectively remove complex information in the entire spectral band and pick out IMEM walnuts. Subsequently, a second test was performed to distinguish between IMEN and INEN walnuts. When the DT was established based on the original bottom spectrum, the CA of IMEN walnuts was 78.6%. The results demonstrate that walnuts with mildew can be identified using this technology, which provides a theoretical basis for the design of machines.

## Figures and Tables

**Figure 1 molecules-27-06776-f001:**
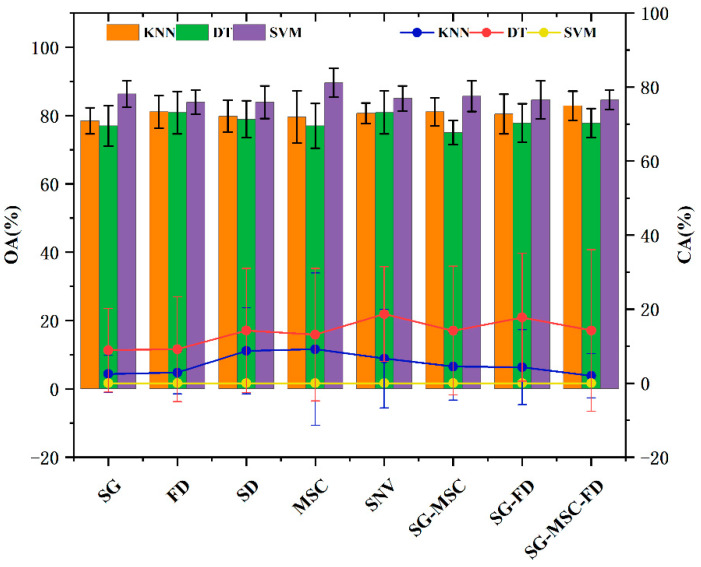
Analysis of the OA and CA of eight pretreatment methods.

**Figure 2 molecules-27-06776-f002:**
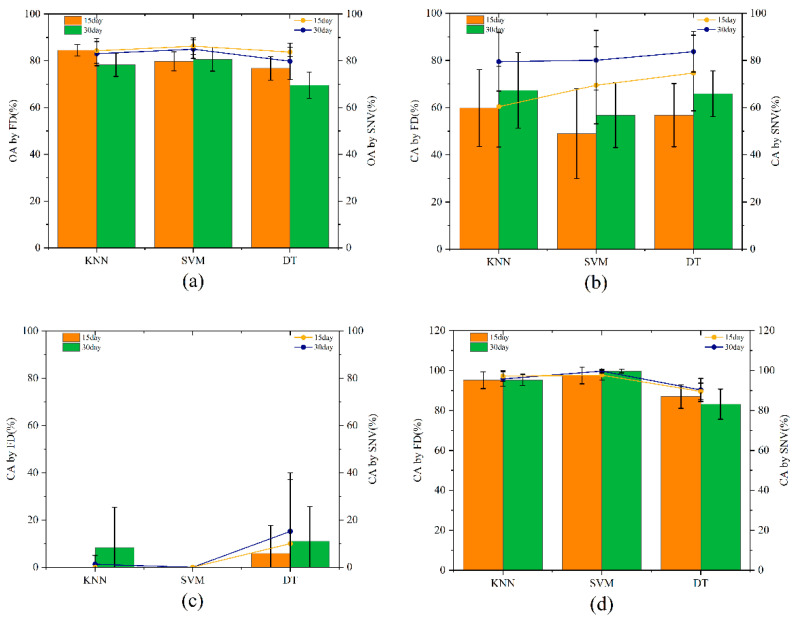
OA (**a**) and CA (**b**–**d**) modeled based on SPA.

**Figure 3 molecules-27-06776-f003:**
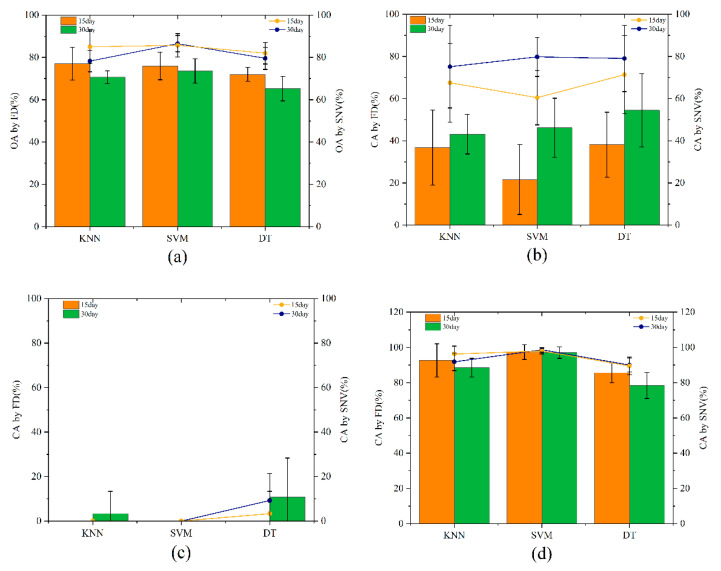
OA (**a**) and CA (**b**–**d**) modeled based on CC.

**Figure 4 molecules-27-06776-f004:**
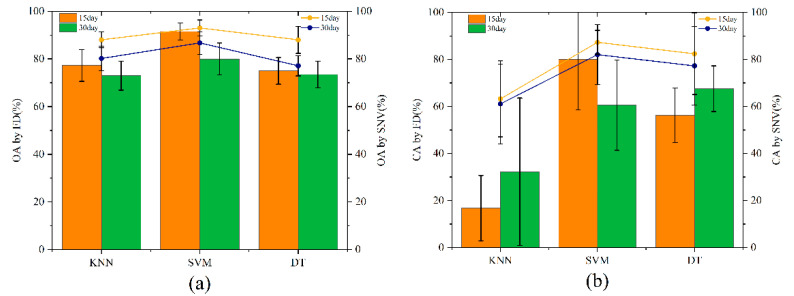
OA (**a**) and CA (**b**–**d**) modeled based on PCA.

**Figure 5 molecules-27-06776-f005:**
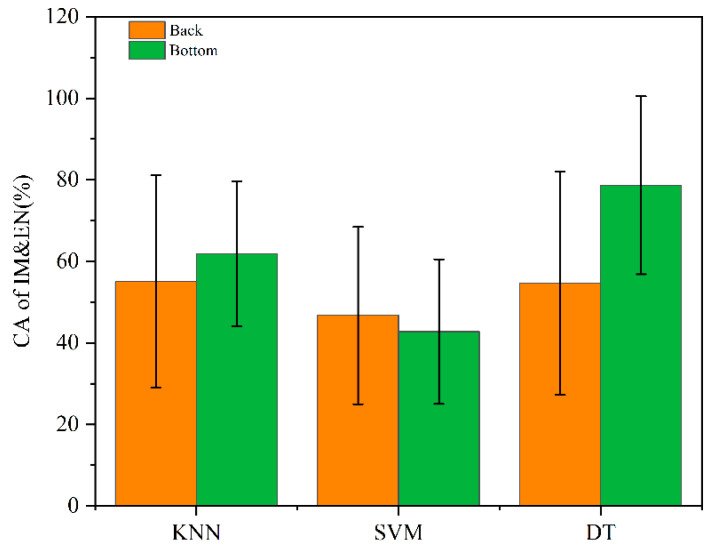
CA of IMEN walnuts.

**Figure 6 molecules-27-06776-f006:**
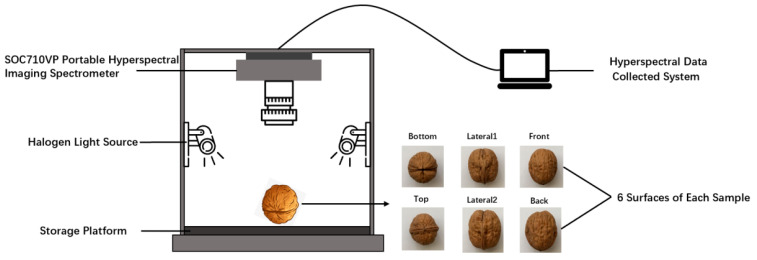
Hyperspectral data acquisition device.

**Figure 7 molecules-27-06776-f007:**
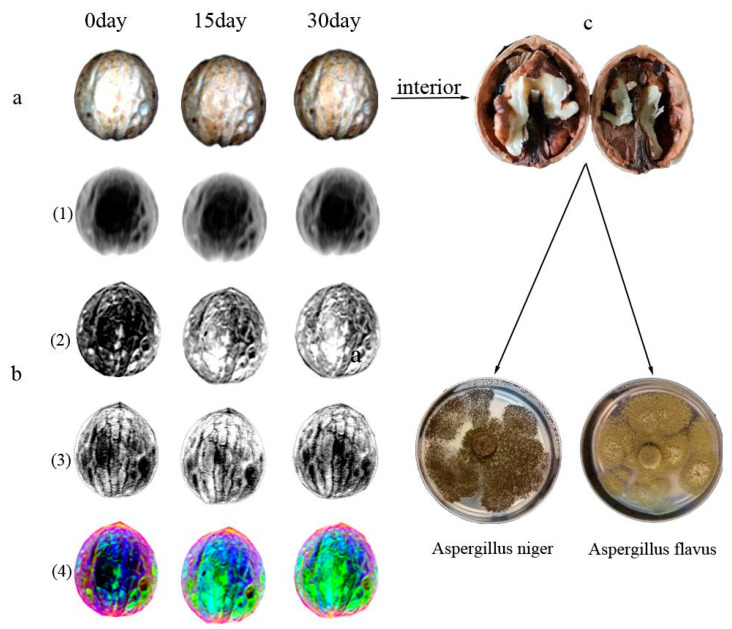
(**a**) Original hyperspectral images of a single walnut sample (R: 638 nm, G:549 nm, B: 456 nm); (**b**) (1)–(3) Score images of PC1–PC3 and (4) pseudo-color images; (**c**) Interior situation of the sample.

**Figure 8 molecules-27-06776-f008:**
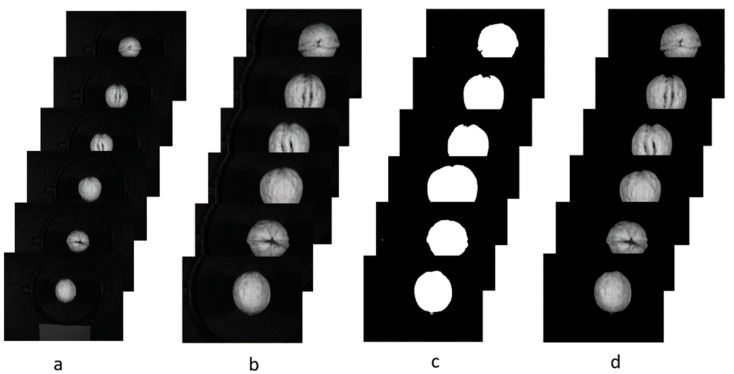
Process flow for extracting reflected spectra: (**a**) the original images, (**b**) cropping, (**c**) threshold method to extract ROI, (**d**) mask.

**Figure 9 molecules-27-06776-f009:**
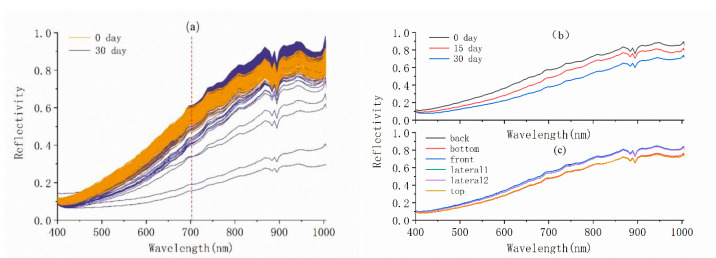
(**a**) Original spectral curves on day 0 and day 30 for all samples; (**b**) Original spectral curves for a single sample at different times; (**c**) Original spectral curves of six surfaces of a single sample.

**Table 1 molecules-27-06776-t001:** OA (%) for different classification models for different mildew situations.

Classification Models	Back		Bottom	
	Day 15	Day 30	Day 15	Day 30
KNN	85.5	74.99	86.65	68.25
DT	85.75	81.95	86.91	68.25
SVM	88.25	83.69	88.96	75.86

**Table 2 molecules-27-06776-t002:** CA (%) of different classification models for different mildew conditions.

Classification Models		KNN				DT				SVM			
Time Input		IM EM	IN EM	IM EN	IN EN	IM EM	IN EM	IM EN	IN EN	IM EM	IN EM	IM EN	IN EN
day 15	Back	62.81	0	10	93.84	86.69	0	6.66	91.06	69.88	0	0	97.84
	Bottom	76.96	0	5	96.13	76.8	0	10	89.78	75.34	0	0	96.73
day 30	Back	78.28	0	3.3	91.15	85.62	0	38.33	88.98	77.93	0	0	100
	Bottom	58.64	0	14.83	88.71	55.55	0	27.83	77.43	50.35	0	0	92.35

## Data Availability

Data are available from the authors on request.

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
