# Peer review of "Non-Destructive Detection of Moldy Walnuts Based on Hyperspectral Imaging Technology"

_molecules, 2022, doi:10.3390/molecules27206776_

Round 1
Reviewer 1 Report
General comments:
The topic of this article is interesting. However, some research results are poor, especially the accuracy 38.33% of the M&EN(internal mildewed and external normal). The result for the M&EN is very important as the authors said, unfortunately, the author did not solved this problem that I am very concerned about.
Moreover, as the authors knew, the size of four category should also be larger, which is an important factor affecting results. The results of current research need to be improved, although the authors used more data processing algorithms. Even with this in mind, please add more experimental details, such as more experimental days, etc..
The reviewer thought that it was somewhat insufficient to select samples in this study with a large size, smooth shell surface and good appearance quality. Why? Please provide more information. Only by expanding the selection range of samples can the established models be used in application.
The paper must need a lot of revisions and also checking by an English speaker.
Reviewer 2 Report
In this manuscript, the authors applied the hyperspectral imaging technology as a non-destructive detection approach to the distinguish the moldy walnuts, with a good accuracy. This topic is interesting and has practical value for food storage. However, this work is not well prepared and lacks in-depth discussion, major revision is needed for paper resubmission. Other comments are listed as below:
1. It seems like that this hyperspectral imaging technology is a qualitative technique. How to quantificationally show the specific mildew degree of walnuts? What is the detection range and limit of this method?
2. The authors should give more information about the mechanism for detection of mildew walnut. What are the possible interference factors and substances?
3. Figure 1 to Figure 4 should have error bars.
4. What about the repeatability and stability of this detection method?
5. The parts of authors’ addresses and acknowledgement are lacked.
6. There are many language errors in the manuscript. For example, in abstract, “Walnut with shell as a popular agricultural product in China” lacks a verb. In introduction, “Chain” should be revised as “China”. There should be the full name when the abbreviation was firstly used. The language of this manuscript should be carefully checked by native speakers.
Reviewer 3 Report
This manuscript proposes the utilization of hyperspectral imaging for the identification of the internal state of whole walnuts with non-destructive testing. Apparently, there is a problem of walnut mildew in China which leads to the loss of significant harvest of this commodity. Until recently, the classification and separation of good walnuts from mildewed walnuts was only carried out manually. Through the proposed technology, this whole process may be automated. While the idea itself is very promising, I am afraid that the manuscript is neither well written, nor well organized.
The Introduction section paves the way in an acceptable manner for the reader to understand the context and the background of the topic. This is supplied with adequate and mostly recent references. However, on reading this section it becomes very clear early on that there needs to be a very thorough language editing to this manuscript. Multiple spelling and grammar mistakes are encountered, making the reading and comprehension a rather hard task. Furthermore, multiple acronyms are written, where their full meaning is either never provided or provided later in the manuscript.
The second section of the manuscript is titled, “Results and Discussion”. This is very strange, since the proposed methodology is not yet fully described.
The classification models, KNN, DT and SVM are not described in the manuscript. The reader is expected to be well aware of what those are. This could have been acceptable if the manuscript was submitted to a journal specifically catering to remote sensing and hyperspectral audience, but I do not think that “MDPI Molecules” is such a journal. Description of the classification models should be provided, at least in brief, where the reader could then be directed to further readings through some related references.
While the references are mostly recent and adequate, their list is not well written, with some references missing dates or have repeated information.
It is my recommendation that this manuscript be returned to the authors with a “Major Changes” requirement. First and foremost for its poor language and organization.
Round 2
Reviewer 1 Report
The authors has addressed some principal problems in the paper, also the entire paper has been thoroughly re-edited. This study as a preliminary exploratory investigation can be published.
In addition, I expect the authors to perform further research and contribute more valuable scientific research information and application ability of this study.
Reviewer 2 Report
The revised version of this manuscript can be accepted in the present form.
Reviewer 3 Report
I am satisfied with the updated version of the manuscript and I believe that it is now ready for publication. Of course, after removing all the editing notes from the PDF.